# Prompt-enhanced Federated Content Representation Learning for Cross-domain Recommendation

## ABSTRACT

Cross-domain Recommendation (CDR) as one of the effective techniques in alleviating the data sparsity issues has been widely studied in recent years. However, previous works may cause domain privacy leakage since they necessitate the aggregation of diverse domain data into a centralized server during the training process. Though several studies have conducted privacy preserving CDR via Federated Learning (FL), they still have the following limitations: 1) They need to upload users' personal information to the central server, posing the risk of leaking user privacy. 2) Existing federated methods mainly rely on atomic item IDs to represent items, which prevents them from modeling items in a unified feature space, increasing the challenge of knowledge transfer among domains. 3) They are all based on the premise of knowing overlapped users between domains, which proves impractical in real-world applications. To address the above limitations, we focus on Privacy-preserving Cross-domain Recommendation (PCDR) and propose PFCR as our solution. For Limitation 1, we develop a FL schema by exclusively utilizing users' interactions with local clients and devising a Local Differential Privacy (LDP) method for gradient encryption. For Limitation 2, we model items in a universal feature space by their description texts. For Limitation 3, we initially learn federated content representations, harnessing the generality of natural language to establish bridges between domains. Subsequently, we craft two prompt fine-tuning strategies to tailor the pre-trained model to the target domain. Extensive experiments conducted on two real-world datasets consistently demonstrate the superiority of our PFCR method compared to the SOTA approaches.

## CCS CONCEPTS

• Information systems → Recommender systems.

## KEYWORDS

Cross-domain Recommendation, Content Representation, Federated Learning

**ACM Reference Format:**
Anonymous Author(s). xxx. Prompt-enhanced Federated Content Representation Learning for Cross-domain Recommendation. In *Proceedings of Make sure to enter the correct conference title from your rights confirmation emai (Conference acronym 'XX)*. ACM, New York, NY, USA, 10 pages. https://doi.org/10.1145/0000000.000000

## 1 INTRODUCTION

Cross-domain Recommendation (CDR) is a key area of recommender systems that aims to improve recommendation quality by leveraging knowledge from multiple domains. It relies on shared patterns and latent correlations, extending recommendations beyond single domains. Techniques such as domain adaptation and transfer learning play a vital role in information migration [7, 8, 17], and have achieved great success in attaining distinguished recommendations. However, a primary limitation of existing CDR methods is that they may leak domain privacy because of the centralized training schema. Direct data aggregation proves infeasible due to safeguards protecting trade secrets, exemplified by regulations such as the General Data Protection Regulation (GDPR) [36]. Furthermore, prior studies grapple with another limitation by aligning domains through direct utilization of users' identity information, under the assumption of overlap. This approach is also unfeasible in many real-world CDR scenarios, chiefly due to user privacy concerns. For instance, users registering for personal services on one platform typically harbor reservations about exposing their identity to other platforms.

Recently, several studies have been focused on conducting privacy preserving in CDR tasks [1, 2, 23]. For instance, Mai et al. [23] propose a federated GNN-based recommender system with random projection to prevent de-anonymization attacks, and a ternary quantization strategy to avoid user privacy leakage. However, previous methods solving PCDR still suffer from the following limitations: 1) They need to upload users' personal information, such as user embedding or user-related model parameters, to the central server. Despite the encryption of user information, there remains a potential for divulging users' private data, as achieving a balance between information encryption and method efficacy is challenging. 2) Existing federated methods predominantly rely on atomic IDs for item modeling, making it challenging to learn unified item representations. The uniqueness of items in different domains and the non-IID nature of data distributions pose difficulties in modeling items within a unified feature space. This limitation impedes the acquisition of universal item representations and hampers knowledge transfer across domains. 3) These methods are premised on the assumption of knowing overlapping users across domains, enabling domain alignment and CDR. However, this assumption carries the risk of serious privacy breaches, as it necessitates the use of users' identity information. Furthermore, identifying common users between domains can expose individuals to de-anonymization attacks, as organizations may exploit this information to infer user preferences in other domains.

To address these limitations, we target Domain-level Privacy-preserving Cross-domain Sequential Recommendation (DPCSR) and propose a Prompt-enhanced Federated Content Representation (PFCR) paradigm as our solution. We consider the sequential characteristic in PCDR since it is a common practice to organize

users' behaviors into sequences. Specifically, to mitigate **Limitation 1**, we propose a federated content representation learning schema, treating domains as clients, with a central server responsible for parameter updates (FedAvg [24] is applied). In PFCR, the gradients related to content representations can be shared across domains (the LDP strategy is also applied), while the user-related gradients are strictly prohibited to prevent privacy leakage. To deal with **Limitation 2**, we model items as language representations (i.e., *semantic ID*) by the associated description text of them so as to learn universal item representations, where the natural language plays the role of a general semantic bridging different domains. Compared with *atomic item ID*, *semantic ID* enables us to represent different domain items in the same semantic space and simultaneously avoids the huge memory and storage footprint caused by the huge number of items in modern recommender systems. To tackle **Limitation 3**, we initially pre-train the federated content representations to fuse non-overlapped domains by leveraging the generality of natural languages, where a global *code embedding table* under a universal semantic space is learned. Subsequently, to adapt the pre-learned knowledge to specific domains, we fine-tune the pre-trained content representations and model parameters with two kinds of prompting strategies.

The main contributions of this work can be summarized as:

- We target DPCSR and solve it by proposing PFCR, where a federated content representation learning schema and a prompt-enhanced fine-tuning paradigm are developed for domain transfer under the non-overlapping scenario.
- We model items in different domains as vector-quantified representations on the basis of their associated description texts, so as to unify them in the same semantic space.
- We develop a federated content representation learning framework for PCDR in the non-overlapping scenario by leveraging the generality of natural languages.
- We design two prompting strategies, namely full prompting, and light prompting, to adapt the pre-learned domain knowledge to the target domain.
- We conduct extensive experiments on two real-world datasets, and the experimental results consistently demonstrate the superiority of PFCR compared with other SOTA methods.

## 2 RELATED WORK

### 2.1 Federated Cross-domain Recommendation

Existing Federated Cross-domain Recommendation (FCDR) studies can be categorized into Cross-Silo Federated Recommendation (CSFR) and Cross-User Federated Recommendation (CUFR) methods according to the nature of clients. CSFR refers to FCDR among organizations and focus on preserving domain-level privacy [2, 3, 21, 23, 37, 47]. For example, Wan et al. [37] devise a privacy-preserving double distillation framework (FedPDD) for CSFR to solve the limited overlapping user issue, which exploits a double distillation strategy to learn both explicit and implicit knowledge, and an offline training schema to prevent privacy leakage. In CUFR, each user is served as a client and tends to conduct user-level privacy preserving by FCDR [25, 46]. For instance, Yan et al. [46] train a general recommendation model on each user's personal device to avoid the leakage of user privacy and devise

an embedding transformation mechanism on the server side for knowledge transfer. However, existing studies mainly rely on all or part of the overlapped users for CDR, and cannot be applied to scenarios in which users are non-overlapped across domains. Our solution falls into the CSFR category and tends to solve DPCSR by proposing a FL framework with federated content representations.

### 2.2 Recommendation with Item Text

Recently, several recommendation methods have been focused on leveraging the content information to represent items to explore the generality of natural languages. Depending on whether text representation is directly used for recommendations, existing studies can be categorized into text representation [12, 19, 26, 31] and code representation-based methods [11, 28]. For example, Hou et al. [12] design a text representation-based method in universal sequence representation approach (UnisRec) by utilizing a Pre-trained Language Models (PLM), where the semantic item encoding is obtained and participates in sentence modeling. But this kind of method is too strict in binding item text and its representation, causing the model to pay much attention to the text features. To address this, Hou et al. [11] first convert the text of the item into a series of distinct indices called "item codes", and then learn these Vector-Quantized (VQ) item representations by engaging with these codes. However, the above methods are all based on a centralized training schema, and none of them consider the generality of contents in helping FCDR.

### 2.3 Recommendation with Prompt tuning

Prompt tuning, initially introduced in the field of NLP, involves designing specific input and output formats to guide PLM in performing specific tasks. Recent researchers have also used prompt tuning to solve the cold-start [30, 43, 44] and cross-domain [9, 40] issues in recommender systems. For example, Wu et al. [44] devise a personalized prompt-based recommendation framework for cold-start recommendation, which builds a soft prompt via a prompt generator based on user profiles, and enables a sufficient training via prompt-oriented contrastive learning. Wang et al. [40] propose a prompt-enhanced paradigm for multi-target CDR, where a unified recommendation model is first pre-trained using data from all the domains, then the prompt tuning process is conducted to capture the distinctions among various domains and users. Though the prompt tuning methods have been widely studied, they are mainly utilized for domain adaption or zero-shot issues, and few of them focus on solving the FCDR task, which is one of the main purposes of this work.

## 3 METHODOLOGIES

### 3.1 Preliminaries

Suppose we have two domains A and B. Let $\mathcal{U}^A = \{u_1^A, u_2^A, \ldots, u_{m_A}^A\}$ and $\mathcal{U}^B = \{u_1^B, u_2^B, \ldots, u_{m_B}^B\}$ be the user sets, $\mathcal{A} = \{A_1, A_2, \ldots, A_{M_A}\}$ and $\mathcal{B} = \{B_1, B_2, \ldots, B_{M_B}\}$ be the item sets in domains A and B, respectively, where $m_A$, $m_B$, $M_A$ and $M_B$ are the corresponding user number and item number in each domain. Each item $A_i \in \mathcal{A}$ (or $B_j \in \mathcal{B}$) is identified by a unique item ID and associated with a description text (such as the product title, introduction, and brand).

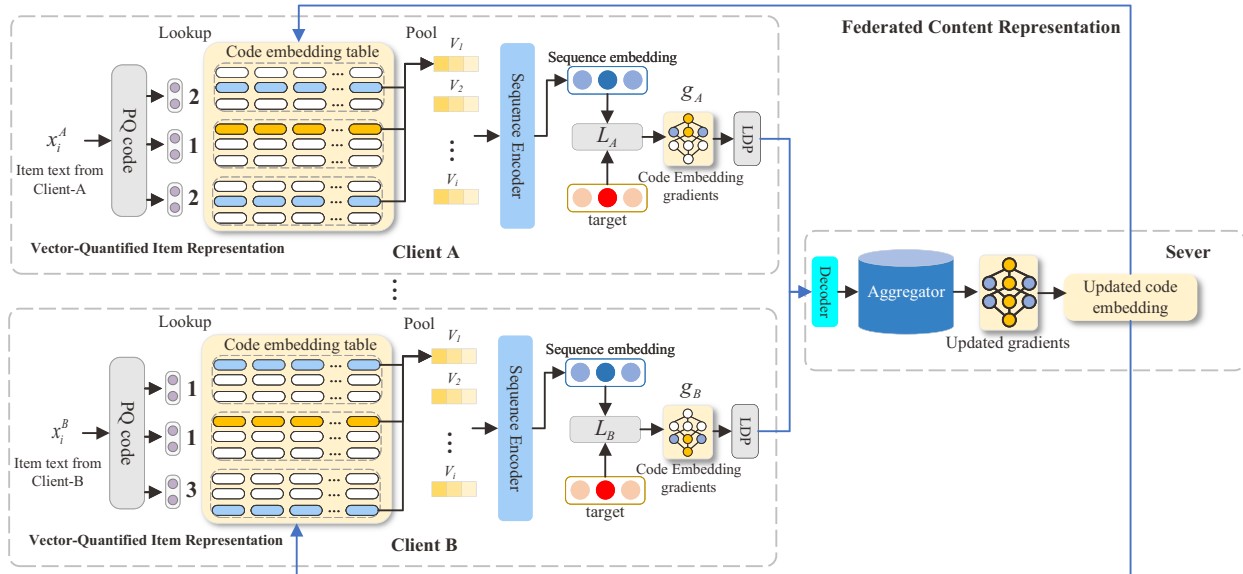

**Figure 1: The system architecture of PFCR in the pre-training stage. In PFCR, the *code embedding table* is pre-learned federally. The orange color within it denotes the index embeddings shared by all the clients, and the blue color indicates the index embedding that only appears in the current client.**

Users can express their preferences by interacting with specific items. Take user $u_i^A$ as an example, we record her sequential behaviors on items in domain A as $\mathcal{S}_i^A = \{A_1, A_2, \ldots, A_j, \ldots\}$. The description text of item $A_i$ is denoted as $\mathcal{T}_i = \{w_i, w_2, \ldots, w_c\}$, where $w_j$ is the content word in natural languages, and $c$ is the truncated length of item text. To represent items in a unified feature space, we share the language vocabulary in both domains. Compared with other ID-based traditional recommendation methods [10, 15, 34], we only take item IDs as auxiliary information, and they will not be used for domain knowledge transfer. Instead, we represent items by deriving generalizable ID-agnostic representations from their description texts. Moreover, although users may simultaneously interact with items in multiple domains or platforms, we do not align them between domains, since it may compromise users' privacy. That is, we assume users and items are entirely non-overlapped in our setting. In this work, we tend to preserve domain privacy in a federated training schema and transfer domain knowledge by a pre-train & prompt learning paradigm with the help of the generalized content representations.

### 3.2 Overview of PFCR

**Motivation.** To conduct PCDR, we resort to FL by viewing domains as clients and local data can only be utilized within clients. Moreover, to prevent attackers from inferring user identities from uploaded public information, we only share the item-related gradients with the protection of LDP. To transfer domain information under the non-overlapping and privacy-preserving scenario, we first embed domain information into the distributed item representation in the pre-training stage, and then adapt the prompts in the fine-tuning stage, so as to meet the specific distribution of the target domain.

The system architecture of PFCR in the pre-training stage is shown in Fig. 1, which is a FL process that consists of Vector-Quantified Item Representation (VQIR), Sequence Encoder (SE), and Federated Content Representation (FCR). VQIR aims at representing items in different domains in the same semantic space. It is the foundation of modeling users' cross-domain personal interests. By unifying domains into the same language feature space, we are able to effectively integrate domain information (see Section 3.3). But as we focus on PCDR, we do not follow the traditional centralized training schema. On the contrary, we devise a framework with the help of the federated content representations (see Section 3.4). In SE, we apply a transformer-style neural network to learn users' sequential interests in each client (see Section 3.4.1). The overall framework of PFCR in the fine-tuning stage is shown in Fig. 2. Two prompting strategies, i.e., full prompting and light prompting, are developed in this phase. In the full prompting strategy (as shown in Fig. 2 (a)), we explore the domain prompts and user prompts for fine-tuning, while in the light prompt learning (as shown in Fig. 2 (b)), only the domain prompts are reserved. In our design, the domain prompts are shared by all the users in the same domain, and the user prompts are related to specific users.

### 3.3 Vector-Quantified Item Representation

As natural language is a universal way to represent items in different domains, it is intuitive to leverage the generality of natural language texts to bridge domain gaps, since similar items will have similar contents even if they are not in the same domain. To model the common semantic information across different domains, we unify items in the same semantic feature space and take the learned text encodings via PLMs as universal item representations. But such a "*text -> representation*" paradigm [12, 19] is too tight in binding

item text and their representations, making the recommender might overemphasize the effect of text features. Moreover, the text encodings from different domains cannot naturally align in a unified semantic space. To address the above challenges, we exploit a "*text -> code -> representation*" schema [11, 28], which first maps item text into a vector of discrete indices (called *item code*), and then employs these indices to lookup the *code embedding table* for deriving item representations. But different from existing studies that learn it in a centralized server, we tend to develop a FL schema for privacy-preserving purposes (the details can be seen in Section 3.4). In this work, we deem the description texts of items as public data and the users' interaction behaviors as privacy information.

*3.3.1 Discrete Item Code Leaning.* To obtain the discrete codes of items, we first encode their description texts into text encodings via PLMs to leverage the generality of natural language text. Then, we map the text encodings into discrete codes based on the optimized product quantization method (the Product Quantization (PQ) [13] algorithm is utilized).

For item text encoding, we utilize the widely used BERT model [4] as the text encoder (the Huggingface model is exploited[1]). Specifically, for a given item $i$, we first insert a [CLS] token at the beginning of its description text $\mathcal{T}_i = \{w_1, \ldots, w_c\}$ and subsequently feed it into BERT to obtain its textual encoding:

$$x_i = \text{BERT}([[\text{CLS}; w_1; \ldots; w_c]]), \quad (1)$$

where [;] represents the concatenation operation, $x_i \in \mathbb{R}^{d_W}$ is the representation of the given text, which is defined as the final hidden vector of the special input token [CLS].

To map the text encoding $x_i$ to discrete codes, the PQ method is applied. PQ defines $D$ sets of vectors, within which each vector corresponds to $M_c$ centroid embeddings with dimension $d_W/D$. Let $a_{k,j} \in \mathbb{R}^{d_W/D}$ be the $j$-th centroid embedding for the $k$-th vector set. In the PQ method, the text encoding vector $x_i$ is first split into $D$ sub-vectors $x_i = [x_{i,1}; \ldots; x_{i,D}]$. Then, for the $k$-th sub-vector of $x_i$, PQ selects the index of its nearest centroid embedding from the corresponding set to generate the discrete code of $x_{i,k}$. The selected index of $x_{i,k}$ can be defined as:

$$c_{i,k} = \arg\min_j \|x_{i,k} - a_{k,j}\|^2 \in \{1, 2, \ldots, M_c\}, \quad (2)$$

where $c_{i,k}$ is $k$-th dimension of the discrete code vector for item $i$.

*3.3.2 Item Code Representation.* Given the discrete item codes (e.g., $(c_{i,1}, \ldots, c_{i,D})$ of item $i$), we can derive item representations by directly performing lookup operation on a *code embedding table* with average pooling.

**Code Embedding Table.** Let $E \in \mathbb{R}^{D \times M_c \times d_V}$ be the global *code embedding table*, where $d_v$ denotes the dimension of the item embedding. There are $D$ code embedding matrices within $E$, and each of them $E^{(k)} \in \mathbb{R}^{M_c \times d_V}$ is shared by the discrete codes of all the items (even if they are not in the same domain). This characteristic allows us to align different domains and embed the common domain information into item embeddings. Moreover, as we share the code embedding among all domains, we can represent items in the same code space, which forms the prerequisite for our subsequent FL endeavors. It is worth noting that we need to let the *code embedding*

[1]https://huggingface.co/bert-base-uncased

*table* in all the clients and servers have the same initialization value to ensure they have the same update direction.

**Lookup Operation.** By performing the lookup operation on $E$, the code embeddings for item $i$ can be denoted as $\{e_{1,c_{i,1}}, \ldots, e_{D,c_{i,D}}\}$, where $e_{k,c_{i,k}} \in \mathbb{R}^{d_V}$ is the $c_{i,k}$-th row of matrix $E^{(k)}$.

Then, we can arrive at the final item representation of item $i$ by conducting the average pooling on the code embeddings:

$$v_i = \text{Pool}\left(\left[e_{1,c_{i,1}}; \ldots; e_{D,c_{i,D}}\right]\right), \quad (3)$$

where $v_i \in \mathbb{R}^{d_V}$ is the final item representation, and $\text{Pool}(\cdot) : \mathbb{R}^{D \times d_V} \to \mathbb{R}^{d_V}$ is the mean pooling method on the $D$ dimension.

## 3.4 Federated Content Representation

Since we focus on the PCDR task, we do not follow the traditional centralized training schema for item representation learning. Instead, we resort to FL, where domains are viewed as clients, and the privacy of user data is strictly utilized only on local clients. To this end, a federated content representation learning paradigm is devised, which involves local training, uploading gradient, and gradient aggregation and synchronization.

*3.4.1 Local Training.* To learn users' sequential interests, we first feed items' VQ representations $\{v_1, v_2, \ldots, v_i, \ldots, v_n\}$ to a transformer-style sequence encoder.

**Sequence Encoder**. It mainly consists of a multi-head self-attention layer (called MH) and a position-aware feed-forward neural network (called FFN) to model items' sequential dependencies. More formally, for each input item representation $v_i$, we first add it to the corresponding position embedding $p_j$ ($j$ is the position of item $i$ in the sequence).

$$h_j^0 = v_i + p_j. \quad (4)$$

Then, we feed $h_j^0$ to MH [35] and FFN [35] to conduct non-linear transformations. The encoding process is defined as follows:

$$H^l = [h_0^l; \ldots; h_n^l], \quad (5)$$

$$H^{l+1} = \text{FFN}\left(\text{MH}\left(H^l\right)\right), l \in \{1, 2, \ldots, L\}, \quad (6)$$

where $H^l \in \mathbb{R}^{n \times d_V}$ denotes the hidden representation of each sequence in the $l$-th layer, $L$ is the total layer number. We take the hidden state $h_i^A = h_n^L$ at the $n$-th position as the sequence representation ($\mathcal{S}_A$ is the input sequence in domain A).

**Optimization Objective**. Given the input sequence $\mathcal{S}_i^A$, we define the next item prediction probability in domain A as follows:

$$P(A_{i+1}|\mathcal{S}_i^A) = \text{Softmax}(h_i^A \cdot V_{\mathcal{A}}), \quad (7)$$

where $V_{\mathcal{A}}$ is the representation of all the items in domain A.

We exploit the cross-entropy loss on all the domains as the local training objective:

$$L_A = -\frac{1}{|\mathcal{S}_A|} \sum_{\mathcal{S}_i^A \in \mathcal{S}_A} \log P(A_{i+1}|\mathcal{S}_i^A), \quad (8)$$

where $\mathcal{S}_A$ is the training set in domain A.

*3.4.2 Gradient Uploading and the Encryption Strategy.* To enhance the local training process and enable the item representations to embed cross-domain information, we need to leverage the user preference information, such as users' interactions, in other domains. But as the privacy leakage concerns, we do not directly upload the user interaction data to the central server. Instead, we only upload model parameters' gradients for aggregation (the details can seen in Section 3.4.3). Then, the accumulated gradients will be passed back to clients to let them engage in the local training. In this distributed learning way, we can embed domain knowledge into pre-trained models, with which we can further conduct CDR.

However, as user-related gradients also have privacy leakage issues (attackers can obtain privacy features from model parameters or gradient through attach methods such as DLG [50]),we only upload item-related gradients in each client (i.e., the *code embedding table*) to the server for accumulation, and prohibit all the the user-related parameters. The uploaded gradients of the *code embedding table* are represented by $g_A$ and $g_B$ in domains $A$ and $B$, respectively.

**Encryption Strategy.** To prevent malicious actors from intercepting these gradients and then using them to infer item information, we further devise a LDP encryption method on these gradients. Traditional LDP methods [14, 22, 27, 42] mainly add Gaussian or Laplace noise to the gradients. But as they may significantly distort the gradient direction, few recent methods exploit the quantization [29, 32, 33] or randomized response [6, 39] methods. However, they can only solve the problem of third-party attacks or untrustworthy partners, since only a single LDP solution is applied. To simultaneously consider both attack types, we propose a composite LDP method on $g_A$ and $g_B$, which consist of a quantization and a randomized response component.

*Quantization.* This component aims to map gradients to a finite number of discrete values to avoid third-party attacks, as the attackers cannot restore the gradient values without knowing the mapping method, even if they can intercept the uploaded gradients. For each element of $g_i^A$ in client A (take domain A as an example), we first clip $g_i^A$ to a certain range $[-\tau, \tau]$ according to its threshold:

$$g_i^A = \text{clamp}(g_i^A, -\tau, \tau), \qquad (9)$$

where $\tau$ is the gradient threshold. Note that, we do not follow previous methods [23, 38] that clip gradients to $[0, 1]$, as gradients usually have richer values, and put them into a smaller interval will cause many gradient values to go to 0.

Then, we scale $g_i^A$ by the following mapping function:

$$q_i^A = \text{round}(\frac{g_i^A + \tau}{s}), \qquad (10)$$

where $\text{round}(\cdot)$ means rounding each element to its nearest integer. $q_i^A \in \{0, 1, \ldots, b-1\}$ is the quantized gradient element. $s = \frac{2\tau}{b}$ is the scaling factor, $b = 2^k$ is the number of quantization buckets, $k$ is the number of quantization bits.

*Randomized response.* To enable our method can also protect against attacks from untrusted partners, the randomized response method [41] is further applied. This method achieves protection by introducing more uncertainty via randomly flipping or displacing the quantized gradient so that its value can be randomly perturbed.

The random noise adding process is defined as:

$$r_i^A = (q_i^A + \text{noise})\%b, \qquad (11)$$

where $\text{noise} \sim \text{Uniform}(0, b)$ is the random noise that follows an Uniform distribution.

Then, to proceed with the flip operation, we generate a flip variable following the Bernoulli distribution. It determines whether the data should undergo the flip operation:

$$c_i^A \sim \text{Bernoulli}(p), p = \frac{e^\epsilon + 1}{e^\epsilon + 2}, q = \frac{1}{e^\epsilon + 2}, \qquad (12)$$

where $p$ and $q = 1 - p$ are the probability parameters, $\epsilon$ is the privacy parameter. Once $c_i^A$ is obtained, the permutation operation is performed on the noisy gradient $r_i^A$:

$$r_i^A = (r_i^A - c_i^A \cdot \text{noise})\%b. \qquad (13)$$

*3.4.3 Gradient Aggregation and Synchronization.* To learn the global *code embedding table* from distributed clients, the gradient aggregation operation is then applied on the server side. However, due to the LDP encryption method, the aggregated gradients may encompass inherent uncertainties and deviation. Therefore, the server needs to undertake further rectification, decode, and reconcile steps on the encrypted gradients.

To ensure each element in the gradient can be processed in the same way, we start by flattening the gradients from both clients, followed by a concatenation operation:

$$r = \text{flatten}(r^A) \oplus \text{flatten}(r^B), \qquad (14)$$

where $r \in \mathbb{R}^{n_c \times d_f}$ is the flattened gradient, $n_c$ is the client number, $d_f = M_c \times d_V$ is the dimension of $r$, $\oplus$ denotes the concatenation operation. $r^A = \{r_1^A, r_2^A, \ldots, r_{M_A}^A\}$ and $r^B = \{r_1^B, r_2^B, \ldots, r_{M_B}^B\}$ are the gradients of $E$ in clients A and B, respectively. Subsequently, to perform scaling and normalization operations during the denoising and recovery process of the gradients, we construct a constant matrix $E^c$ as follows:

$$E^c = [(p - q)]_{i=1}^{n_c \times d_f}, \qquad (15)$$

where $E^c \in \mathbb{R}^{2 \times d_f}$ is a constant matrix with the same shape as $r$. $p$ and $q$ are employed to introduce the probabilities of inversion and permutation operations. By multiplying this constant matrix with the gradients after random response, each element is effectively subjected to an inverse operation during the decoding and correction process, consequently restoring the denoised gradient information.

We utilize FedAVG to aggregate the gradients, and determine the aggregation weight based on the ratio of the client data to the total data:

$$w_i = \frac{m_i}{\sum_{i=1}^{n_c} m_i}. \qquad (16)$$

The rectification and aggregation process can then be expressed as:

$$g = \sum_{i=1}^{n_c} r_i \cdot E_i^c \cdot w_i. \qquad (17)$$

After that, we reshape the gradients back to their original dimensions and then decode them back to their previous range:

$$g = \text{reshape}(g) \cdot s - \tau. \qquad (18)$$

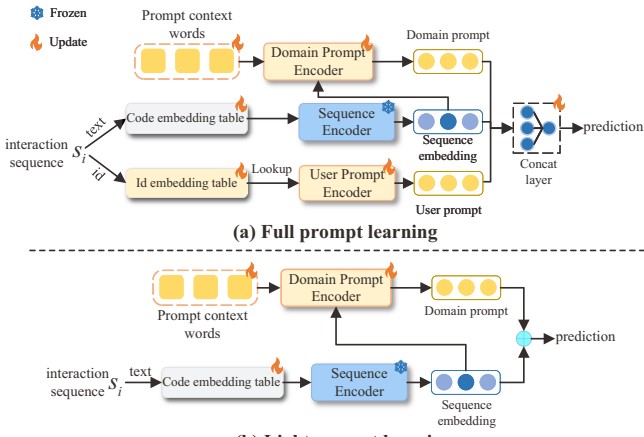

**(a) Full prompt learning**

**(b) Light prompt learning**

**Figure 2: The system architecture of PFCR in the prompt-tuning stage. The components with yellow color are the prompts to be fine-tuned.**

Finally, we utilize the decoded gradients to update the embedding within the server, which can be defined as:

$$E \leftarrow E - \alpha \cdot \boldsymbol{g}. \tag{19}$$

We synchronize this updated embedding to all clients, followed by repeating the aforementioned training process until the pre-training phase convergences. Note that during the initialization phase of FL, we've ensured that the initial random parameters of the embedding on the server match those of the clients. As a result, the updated embedding is valid at this point.

## 3.5 Domain-adaptive Prompting Paradigm

To retrieve the domain knowledge from pre-trained models, we further fine-tune the federated content representation through domain-adaptive prompts, i.e., full prompt and light prompt learning, to enhance CDR.

*3.5.1 The Full Prompting Schema.* In this prompting paradigm, we exploit two kinds of soft prompts, i.e., domain prompt and user prompt, for domain adaption.

**Domain Prompt.** This is to extract the common preferences shared by all the users within each domain, which consists of the prompt context words and a domain prompt encoder. Suppose we have $d_W$ context words in the domain prompt $P_{\text{domain}} \in \mathbb{R}^{d_W \times d_V}$ (we set $d_W$ as the batch size for the convenience of calculation). Then, we encode it by a multi-head attention layer (called MA). But different from the vanilla self-attention, we take the sequence embedding $\boldsymbol{h}$ obtained by the pre-trained model as the queries. This encoding process can be defined as:

$$MA(P_{\text{domain}}) = [\text{head}_1; \text{head}_2; \ldots; \text{head}_{n_h}]\mathbf{W}^O, \tag{20}$$

$$\text{head}_i = \text{Attention}(\boldsymbol{h}_d \mathbf{W}_i^Q, P_{\text{domain}}\mathbf{W}_i^K, P_{\text{domain}}\mathbf{W}_i^V), \tag{21}$$

where $n_h$ denotes the number of heads, $\mathbf{W}_i^Q, \mathbf{W}_i^K, \mathbf{W}_i^V \in \mathbb{R}^{d_V \times d_V/n_h}$, and $\mathbf{W}^O \in \mathbb{R}^{d_V \times d_V}$ are learnable parameters, $\boldsymbol{h}_d$ is the representation of $d_W$ sequences.

**User Prompt.** This is to model users' personal preferences in each domain. We represent it by the sequences of original item IDs, since they are unique to each user and can as supplementary information to user preferences (they are specific to each domain, and do not need to have cross-domain information). Then, we encode it ($P_{\text{user}}$) by a transformer-style user prompt encoder(called UPE), which is defined as:

$$P_{\text{user}} = \text{UPE}(\mathcal{S}). \tag{22}$$

To this end, we concatenate domain prompt, user prompt, and sequence embedding, followed by a fully connected work, to make predictions:

$$\boldsymbol{h}_{\text{full}} = \mathbf{W}^C[P_{\text{domain}} \oplus P_{\text{user}} \oplus \boldsymbol{h}_{\mathcal{S}}] + b^C, \tag{23}$$

where $\mathbf{W}^C : \mathbb{R}^{3d_V} \rightarrow \mathbb{R}^{d_V}$ represents the weight matrix.

*3.5.2 The Light Prompting Paradigm.* To reduce the storage and computational costs in the fine-tuning process, we further consider a light prompting paradigm by removing the item-level features (i.e., the user prompt) and concatenation layer. The final sequence representation in this schema is:

$$\boldsymbol{h}_{\text{light}} = P_{\text{domain}} + \boldsymbol{h}_{\mathcal{S}}. \tag{24}$$

**Learning Objective.** For both paradigms, we minimize the following cross-entropy loss for learning optimal prompts (take domain A as an example):

$$L_A = -\frac{1}{|\mathcal{S}_A|} \sum_{\mathcal{S}_i^A \in \mathcal{S}_A} \log P(A_{i+1}|\mathcal{S}_i^A), \tag{25}$$

where $P(A_{i+1}|\mathcal{S}_i^A) = \text{Softmax}(\boldsymbol{h}_{\text{full}}(\boldsymbol{h}_{\text{light}}) \cdot V_{\mathcal{A}})$ is the probability of predicting the next item $A_{i+1}$. The two-stage optimization algorithm is shown in Appendix B.

## 4 EXPERIMENTAL SETUP

### 4.1 Research Questions

We fully evaluate our PFCR method by answering the following research questions:

**RQ1** How does PFCR perform compared with the state-of-the-art baselines?

**RQ2** How do the key components of PFCR, i.e., Vector-Quantified Item Representation (VQIR), Federated Content Representation (FCR), and Domain-adaptive Prompting (DP), contribute to the performance of PFCR?

**RQ3** How do different federated learning algorithms affect PFCR?

**RQ4** What are the impacts of the key hyper-parameters on the performance of PFCR?

### 4.2 Datasets and Evaluation Protocols

We conduct experiments on two pairs of domains, i.e., "*Office-Arts*", and "*OnlineRetail-Pantry*", in Amazon[2] and OnlineRetail[3] to evaluate our PFCR method. To conduct CDR, we select the Office and Arts domains in Amazon as our learning objective (i.e., the "*Office-Arts*" dataset). To further evaluate PFCR on the cross-platform scenario, we select Pantry as one domain and the data from OnlineRetail

---
[2]https://nijianmo.github.io/amazon/index.html
[3]https://www.kaggle.com/carrie1/ecommerce-data

**Table 1: Comparison results on the *Office-Arts* and *OnlineRetail-Pantry* datasets. PFCR (l) means we use the light prompting strategy for fine-tuning. PFCR (f) means we exploit the full prompting strategy for fine-tuning. We will use PFCR to refer to PFCR (f) in the following unless otherwise specified.**

| Dataset | Metric | ID-Only Methods | | | Cross-Domain Methods | | ID-Text Methods | | Text-Only Methods | | Light | Full |
|---|---|---|---|---|---|---|---|---|---|---|---|---|
| | | GRU4Rec[10] | SASRec[15] | BERT4Rec[34] | CCDR[45] | RecGURU[18] | FDSA[48] | S³Rec[49] | ZESRec[5] | VQRec[11] | PFCR (l) | PFCR (f) |
| Office | Recall@10 | 0.1049 | 0.1157 | 0.0693 | 0.0549 | 0.1145 | 0.1091 | 0.1030 | 0.0641 | 0.1207 | **0.1215*** | 0.1206 |
| | NDCG@10 | 0.0804 | 0.0663 | 0.0554 | 0.0290 | 0.0768 | **0.0821** | 0.0653 | 0.0391 | 0.0742 | 0.0775 | 0.0782 |
| | Recall@50 | 0.1626 | 0.1799 | 0.1182 | 0.1095 | 0.1757 | 0.1697 | 0.1613 | 0.1113 | 0.1934 | **0.1945*** | 0.1938 |
| | NDCG@50 | 0.0929 | 0.0803 | 0.0617 | 0.0401 | 0.0901 | **0.0953** | 0.0780 | 0.0493 | 0.0900 | 0.0933 | 0.0941 |
| Arts | Recall@10 | 0.0893 | 0.1048 | 0.0638 | 0.0671 | 0.1084 | 0.1016 | 0.1003 | 0.0664 | 0.1132 | **0.117*** | 0.1153 |
| | NDCG@10 | 0.0561 | 0.0557 | 0.0364 | 0.0348 | 0.0651 | 0.0671 | 0.0601 | 0.0375 | 0.0624 | 0.0651 | **0.0677*** |
| | Recall@50 | 0.1318 | 0.1948 | 0.1318 | 0.1478 | 0.1979 | 0.1887 | 0.1888 | 0.1323 | 0.2160 | **0.2208*** | 0.2199 |
| | NDCG@50 | 0.0745 | 0.0753 | 0.0510 | 0.0523 | 0.0845 | 0.0860 | 0.0793 | 0.0518 | 0.0848 | 0.0877 | **0.0904*** |
| OR | Recall@10 | 0.1461 | 0.1484 | 0.1392 | 0.1347 | 0.1467 | 0.1487 | 0.1418 | 0.1103 | 0.1496 | 0.1522 | **0.1561*** |
| | NDCG@10 | 0.0715 | 0.0684 | 0.0642 | 0.0658 | 0.0535 | 0.0715 | 0.0654 | 0.0535 | 0.0708 | 0.0710 | **0.0739*** |
| | Recall@50 | 0.3848 | 0.3921 | 0.3668 | 0.3587 | 0.3885 | 0.3748 | 0.3702 | 0.2750 | 0.3900 | 0.3958 | **0.3982*** |
| | NDCG@50 | 0.1235 | 0.1216 | 0.1137 | 0.1108 | 0.1188 | 0.1208 | 0.1154 | 0.0896 | 0.1234 | 0.1241 | **0.1266*** |
| Pantry | Recall@10 | 0.0379 | 0.0467 | 0.0281 | 0.0408 | 0.0469 | 0.0414 | 0.0444 | 0.0454 | 0.0561 | **0.0620*** | 0.0616 |
| | NDCG@10 | 0.0193 | 0.0207 | 0.0142 | 0.0203 | 0.0209 | 0.0218 | 0.0214 | 0.0230 | 0.0251 | 0.0272 | **0.0293*** |
| | Recall@50 | 0.1285 | 0.1298 | 0.0984 | 0.1262 | 0.1269 | 0.1198 | 0.1315 | 0.1141 | 0.1528 | 0.1560 | **0.1591*** |
| | NDCG@50 | 0.0386 | 0.0385 | 0.0292 | 0.0385 | 0.0379 | 0.0385 | 0.0400 | 0.0378 | 0.0458 | 0.0474 | **0.0503*** |

Significant improvements over the best baseline results are marked with * (t-test, $p <$ .05).

**Table 2: Ablation studies on the *Office-Arts* dataset.**

| Variants | Office | | | | Arts | | | |
|---|---|---|---|---|---|---|---|---|
| | Recall | | NDCG | | Recall | | NDCG | |
| | @10 | @50 | @10 | @50 | @10 | @50 | @10 | @50 |
| PFCR-VFD | 0.1157 | 0.1799 | 0.0663 | 0.0803 | 0.1048 | 0.1948 | 0.0557 | 0.0753 |
| PFCR-FD | **0.1207** | 0.1934 | 0.0742 | 0.0900 | 0.1132 | 0.216 | 0.0624 | 0.0848 |
| PFCR-D | 0.1213 | **0.1943** | 0.0771 | 0.0930 | 0.1146 | 0.2176 | 0.0638 | 0.0862 |
| PFCR-F | 0.1169 | 0.1875 | 0.0738 | 0.0892 | 0.1147 | 0.2167 | 0.0655 | 0.0877 |
| **PFCR** | 0.1206 | 0.1938 | **0.0782** | **0.0941** | **0.1153** | **0.2199** | **0.0677** | **0.0904** |

as another (i.e., the "*OnlineRetail-Pantry*" dataset). To conduct sequential recommendations, all the users' interaction behaviors are organized in chronological order. To satisfy the non-overlapping characteristic, all the users and items are disjoint in different domains. For a more detailed description of the datasets, please see Appendix A.

For evaluation, we adopt the leave-one-out strategy and utilize the last three items in each user sequence as the test, validation, and training targets, respectively. We measure the experimental results by the commonly used metrics Recall@$K$ and NDCG@$K$, where $K$ is $\in \{10, 50\}$.

## 4.3 Baselines

We compare PFCR with the following four types of baselines. 1) ID-only sequential recommendation methods: GRU4Rec [10], SASRec [15], and BERT4Rec [34]. 2) Non-overlapping CDR methods: CCDR [45] and RecGURU [18]. 3) ID-Text recommendation methods: FDSA [48] and S³-Rec [49]. 4) Text-Only recommendation methods: ZESRec [5] and VQRec [11]. We do not compare with the overlapping CDR methods since they need an identical number of training samples in both source and target domains. To demonstrate the effectiveness of our federated strategies, we make comparisons with several typical federated methods, and report their results in Table 3.

## 4.4 Implementation Details

We implement PFCR based on the PyTorch framework accelerated by an NVidia RTX 2080 Ti GPU. 1) In the pre-training stage, all the clients utilize the Adam optimizer for local training with learning rate of 0.001 and batch size as 1,024 in both datasets. The head number is set as 4 and the dimension of the hidden state is set as 300 in the sequence encoder. In the federated training process, we choose the model with the highest performance of Recall@10 on the validation dataset and adopt early stopping with patience of 10. For the dimension of the *code embedding table*, we set $M$ =48, $D$=256 for the "*Office-Arts*" dataset, and $M$ =32, $D$=256 for the "*OnlineRetail-Pantry*" dataset. For the LDP method, we search the value of the privacy $\epsilon$ within $\in [0.1, 0.9]$, and the number of quantization buckets $b$ within $\in [2^6, 2^{10}]$ in both datasets. On the server-side, we exploit the FedAVG algorithm for gradient aggregation. The total iterative process is conducted in 10 rounds. 2) In the prompt tuning stage, the head number is set as 4, and the dimension of the hidden state is set as 300 in both domain prompt encoder and user prompt encoder. The number of the context words $d_W$ in the domain prompt is set as 1,024. For the hyper-parameters in the baselines, we set their values based on their publications, and fine-tune them on both datasets.

## 5 EXPERIMENTAL RESULTS (RQ1)

The comparison results are presented in Table 1, from which have the following observations: 1) Our PFCR methods (i.e., PFCR (l) and PFCR (f)) significantly outperforms other SOTA methods in almost all the metrics, demonstrating the effectiveness of our FCR method and the importance of the general content in transferring cross-domain formation in the federated scenario. 2) The improvement of our PFCR methods over ID-only methods (i.e., GRU4Rec, SASRec, and BERT4Rec), indicating the usefulness of our FL framework in conducting PCDR. Our methods have better performance than Text-only methods (i.e., ZESRec and VQRec), showing the

**Table 3: Results of different federated learning strategies on the *Office-Arts* dataset.**

| Methods | Office | | | | Arts | | | |
| --- | --- | --- | --- | --- | --- | --- | --- | --- |
| | Recall | | NDCG | | Recall | | NDCG | |
| | @10 | @50 | @10 | @50 | @10 | @50 | @10 | @50 |
| FedProx [20] | 0.1197 | 0.1912 | 0.0751 | 0.0907 | 0.1132 | **0.2179** | 0.0635 | 0.0862 |
| Scaffold [16] | 0.1146 | 0.1839 | 0.0742 | 0.0893 | 0.1115 | 0.2096 | **0.0695** | **0.0907** |
| Popcode code | 0.1209 | 0.1937 | 0.0742 | 0.0900 | 0.1139 | 0.2153 | 0.0629 | 0.0850 |
| **FedAVG** | **0.1213** | **0.1943** | **0.0771** | **0.0930** | 0.1146 | 0.2176 | 0.0638 | 0.0862 |

benefit of conducting CDR, and our method can effectively transfer domain knowledge in the non-overlapping scenario. 3) The cross-domain methods (i.e., CCDR and RecGURU) do not show impressive improvement over the single domains methods, denoting that non-overlapping CDR is a challenging task since there is no direct information to align domains. Our methods outperform CDR methods, demonstrating the usefulness of the contents in modeling the generality of domain information.

# 6 MODEL ANALYSIS

## 6.1 Ablation Studies (RQ2)

To show the importance of different model components, we further conduct ablation studies by comparing with the following variations of PFCR: 1) PFCR-VFD: This method excludes the VQIR, FCR, and DP components from PFCR. 2) PFCR-FD: This method removes the FCR and DP modules from PFCR. 3) PFCR-D: This method detaches the DP module from PFCR. 4) PFCR-F: This model does not use the FCR component in PFCR. The results of the ablation studies are shown in Table 2, from which we can observe that: 1) PFCR still has the best performance over its other variations. The gap between PFCR and PFCR-VFD indicates the importance of the components (i.e., VQIR, FCR, and DP) in learning the federated content presentations. 2) PFCR-VF performs better than PFCR-VFD, indicating the effectiveness of encoding items by semantic contents. 3) PFCR has a better performance than PFCR-D, showing the usefulness of conducting prompt learning in the DPCSR task. 4) The gap between PFCR and PFCR-F, indicating the importance of our FL strategy.

## 6.2 Impacts of Different Federated Learning Strategies (RQ3)

To show the impact of different FL strategies, we further compare FedAVG (utilized in our PFCR method) with the following methods: FedProx [20], Scaffold [16], and PopCode. PopCode is the method that only aggregates the code gradients with high frequencies in clients (i.e., popular codes). The experimental results are shown in Table 3, from which we can conclude that: 1) FedAVG outperforms FedProx and Scaffold, demonstrating the importance of simultaneously learning the common and domain-specific features of the code embeddings. Paying more attention to the common knowledge across domains (i.e., FedProx and Scaffold) cannot achieve better results. 2) FedAVG performs better than PopCode, indicating the benefit of aggregating all the codes. Only updating a subset of gradients in each client will distort the learning direction of the *code embedding table*, which results in sub-optimal results. 3) Beyond performance, we notice that FedProx and Scaffold upload

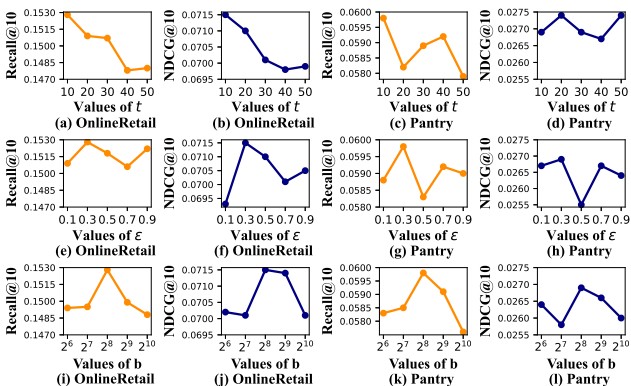

**Figure 3: Impact of the hyper-parameters $t, \epsilon, b$ on the *OnlineRetail-Pantry* dataset.**

more parameters than FedAVG, and have heavier computational costs.

## 6.3 Impact of Hyper-parameters (RQ4)

We explore the impact of three key hyper-parameters, i.e., the number of communication round $t$ in FL, the privacy parameter $\epsilon$ in LDP, and the number of quantified buckets $b$ in the quantization component of LDP. The experimental results are reported in Fig 3. Due to the space limitation, we only show the results on "*OnlineRetail-Pantry*", and similar results are achieved on "*Office-Arts*". From Fig 3, we can observe that: 1) $t$ significantly impacts the performance of PFCR. The value of $t$ is not that the bigger the better. A higher value of $t$ will result in excessive updates, leading to the performance decline. 2) PFCR does not have a definite chaining trend as $\epsilon$ changes since the randomized response method is different from Laplace or Gaussian noise that is added on all the gradients, while ours are only on part of them. 3) A proper value of $b$ is important to PFCR. An over-big value of $b$ will map the gradients to an overlarge range and will result in a performance decline. Similarly, an over-small $b$ will limit the gradients in an over-small range, and let the gradients have less representational ability.

# 7 CONCLUSIONS

In this work, we target DPCSR and propose a PFCR paradigm as our solution with a two-stage training schema (the pre-training and prompt tuning stages). The pre-training phase is dedicated to achieving domain fusion and privacy preservation by harnessing the generality inherent in natural languages. Within this phase, we introduce a federated content representation learning method. The prompt tuning phase is geared towards adapting the pre-learned domain knowledge to the target domain, thereby enhancing CDR. To achieve this, we have devised two prompt learning techniques: the full prompt and light prompt learning methodologies. The experimental results on two real-world datasets demonstrate the superiority of our PFCR method and the effectiveness of our federated content representation learning in solving the non-overlapped PCDR tasks.

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

## A   DATASETS

We conduct experiments on two pairs of domains, i.e., "*Office-Arts*", and "*OnlineRetail-Pantry*", in Amazon and OnlineRetail to evaluate our PFCR method. Amazon is a product review dataset that records users' rating behaviors on products in different domains. OnlineRetail is a UK online retail platform that records users' purchase histories on products within it. In both datasets, each product has a descriptive text, which is used to introduce the product such as function, purpose, etc. To conduct CDR, we select the Office and Arts domains in Amazon as our learning objective (i.e., the "*Office-Arts*" dataset). To further evaluate PFCR on the cross-platform scenario, we select Pantry as one domain and the data from OnlineRetail as another (i.e., the "*OnlineRetail-Pantry*" dataset). To conduct sequential recommendations, all the users' interaction behaviors are organized in chronological order. To satisfy the non-overlapping characteristic, all the users and items are disjoint in different domains. The only connection between domains is that similar products may have similar descriptive texts. To alleviate the impact of sparse data, we filter out users and items with less than 5 interactions. The statistics of our resulting datasets are reported in Table 4. Note that, since we focus on disjoint domains, we do not need the same number of training samples for different domains as the overlapping CDR methods do.

**Table 4: Statistics of the preprocessed datasets. "Avg. $n$" denotes the average length of the user interaction sequence.**

| Datasets | #Users | #Items | #Inters. | Avg. $n$ |
|---|---|---|---|---|
| **Office** | 87,436 | 25,986 | 684,837 | 7.84 |
| **Arts** | 45,486 | 21,019 | 395,150 | 8.69 |
| **OnlineRetail** | 16,520 | 3,469 | 519,906 | 26.90 |
| **Pantry** | 13,101 | 4,898 | 126,962 | 9.69 |

## B   TWO-STAGE TRAINING PROCESS

The training process of PFCR is shown in Algorithm 1, which consists of the pre-training and prompt tuning stages. In the pre-training stage, each client first learns the *code embedding table* and the sequence encoder locally based on users' local behavior data on this domain. At the end of each training round, the accumulated gradients of the code embedding, encrypted with LDP, are uploaded to the server for decoding and aggregation. Then, the server utilizes the aggregated gradients to update the *code embedding table* and

synchronizes it across all clients. In the second stage, each client conducts prompt tuning to adapt the pre-learned domain knowledge to the specific domain by fixing the parameters that are not in the *code embedding table* and prompts.

---

**Algorithm 1:** The two-stage training process of PFCR.

**Input:** Interaction sequence from two clients, $\mathcal{S}^A$ and $\mathcal{S}^B$; descriptions of all items, $\mathcal{T}$.

**Output:** Next-item predictions for each user in the client.

1 **Stage 1: Federated Pre-training:**

2 **Initialization:** Obtain the representation vector $v_i$ for each item using the method described in Section 3.3.;

3 **for** *each epoch i with i = 1, 2, . . .* **do**

4    **Client_Executes:**

5    **for** *each client j* **do**

6       **for** *each batch* **do**

7          Calculate local loss $L_j$ via Eq. (8) ;

8          Accumulate the gradients of code embedding: $g_j = g_j + \nabla L_j(\theta)$ ;

9       **end**

10       Apply the LDP encryption to $g_j$ via Eq. (9) - (13) ;

11       Upload the encrypted gradients to the server ;

12    **end**

13    **Server_Executes:**

14    Decode and aggregate the received client gradients using Eq. (14) - (18) ;

15    Update global code embedding $E$ via Eq. (19) ;

16    Synchronize $E$ to all the clients ;

17 **end**

18 **Stage 2: Prompt Tuning in clients:**

19 **while** *not converge* **do**

20    Freeze all parameters except for the code embedding ;

21    Obtain the domain prompt via Eq. (20) ;

22    Obtain the user prompt via Eq. (22) ;

23    Obtain the final output by combining prompts and sequence output using Eq. (23) or Eq. (24) ;

24    Train using the final output with Eq. (25) ;

25 **end**

---

