# OpenReview forum: "Prompt-enhanced Federated Content Representation Learning for Cross-domain Recommendation"
_ACM.org/TheWebConf/2024/Conference — TheWebConf24_

### Official Review · Reviewer_WPh4 · 2023-11-22

**Novelty:** 4
**Technical Quality:** 5

**Review:**

This research proposes an algorithm, Prompt-enhanced Federated Content Representation (PFCR), for privacy preserving cross-domain recommendations problem where multiple parties have recommendation data but they may or may not have any user or item overlap. The algorithm utilizes federated learning, local differential privacy, vector quantized item text and prompt tuning to accomplish this. In the pre-training stage, the item text embedding is learned as a federated learning problem where each party is a client. A joint global code embedding table is learned through this process where each item is keyed using a vector-quantized id generated through the Product Quantization algorithm. This id allows bridging the domain gap through the use of natural language to describe the item. To learn, noisy gradients using randomized response from each party are applied to update the global embedding with a local differential privacy guarantee. In the second stage of training, the content representation is fine-tuned using prompt encoding. The authors present a full and light prompting paradigm where both domain level and user level fine tuning happens in the full version and only domain level finetuning in the light version.

The authors further conduct a series of detailed experiments to understand how this compares to the state of the art, ablation studies to understand what parts are necessary, how different federated learning algorithms might affect this algorithm and finally the impact of federated learning and differential privacy hyperparameters.

Pros:
- Mostly easy to understand paper that applies recommendations with item text to domain level privacy preserving cross domain recommendations using federated learning.
- Performs clear experiments to understand the algorithm from a variety of perspectives and notes that algorithm outperforms other state of the art baselines.

Cons:
- The experiments regarding hyperparameters seems to not produce any interesting results. Specifically, its weird that even in the high privacy regime with small epsilon has no impact on performance. It might be interesting to see if performance changes as epsilon goes to infinity.
- Can the authors clarify what is their definition of neighboring datasets for their DP guarantee?

**Questions:**

See cons.

**Reviewer Confidence:**

3: The reviewer is confident but not certain that the evaluation is correct

**Scope:**

4: The work is relevant to the Web and to the track, and is of broad interest to the community

---

### Official Review · Reviewer_ruVM · 2023-11-23

**Novelty:** 6
**Technical Quality:** 7

**Review:**

This work studies the privacy-preserving Cross-domain Recommendation (PCDR) task by leveraging the capability of federated content representation learning, where a prompt-enhanced federated content representation paradigm is proposed. Compared with previous solutions, this proposal is advanced in the following aspects: 1) It develops a federated learning schema by exclusively utilizing users' interactions with local clients to address the user privacy leakage issue. 2) It models items by their description texts to unify them in a universal feature space. 3) It solves the non-overlapping issue by the federated content representation learning and two prompt fine-tuning strategies. The authors conduct extensive experiments on two real-world datasets, and the results show the advance of the proposal compared with other SOTA methods. The overall solution sounds reasonable and has technical contributions to the PCDR task.

Strengths:
1. This work studies the PCDR task, which is interesting and may help promote cooperation between organizations.
2. This work addresses three limitations with previous methods by leveraging the capability of federated representation learning and prompt tuning techniques.
3. The overall solution sounds reasonable and has technical contributions to the PCDR task.
4. The authors conduct extensive experiments on two real-world datasets, and the results are impressive compared with the SOTA methods.

Weaknesses:
1. The authors can provide an illustrated example to further explain the motivation that leverages the generality of natural languages to solve the non-overlapping issue for better understanding.
2. As this work is focused on conducting recommendations via model items' descriptional texts, its advance over pure IDs can be further discussed.
3. The authors are suggested to further discuss this study's limitations and application scope, such as the data and scenarios where this method is not applicable.

**Questions:**

Please see Weaknesses 1-3.

**Ethics Review Description:**

N.A.

**Reviewer Confidence:**

4: The reviewer is certain that the evaluation is correct and very familiar with the relevant literature

**Scope:**

4: The work is relevant to the Web and to the track, and is of broad interest to the community

---

### Official Review · Reviewer_1pH6 · 2023-11-23

**Novelty:** 6
**Technical Quality:** 5

**Review:**

Pros
1. The paper introduces a novel prompt-enhanced federated content representation learning approach, which address the three limitations in the CDR scenario ((uploading personal information, reliance on atomic item IDs, and the requirement of knowing overlapped users). The idea is very innovative.
2. This paper is clear in presenting the motivation, limitations, experiments and analysis. Besides, the formulas and figures  in this paper is great.

Cons
1. This paper compares ID-Only Methods, Cross-Domain Methods, ID-Text Methods, and Text-Only Methods in experiments. It’s better to compare some novel federated CDR method to  prove its effectiveness in the field of privacy protection CDR.

**Questions:**

Please see Cons.

**Ethics Review Description:**

NA.

**Reviewer Confidence:**

4: The reviewer is certain that the evaluation is correct and very familiar with the relevant literature

**Scope:**

4: The work is relevant to the Web and to the track, and is of broad interest to the community

---

### Official Review · Reviewer_29f5 · 2023-11-24

**Novelty:** 4
**Technical Quality:** 4

**Review:**

Pros:

1.	The problem is interesting and has a wide application.

2.	The paper is stated with fluent and understandable language.

Cons:

1.	The motivation for this work needs further clarification. In the Introduction on the first page, the statement regarding "relying on atomic IDs" in Limitation 2 might be confusing. Since the feature representation modeling of an item typically relies on user-item interactions or content features rather than solely on "atomic IDs."

2.	Some illustrations are not clear

a)	In the third paragraph of the Introduction part, the mention of "Sequential Recommendation" appears abruptly, since the previous limitations do not mention "Sequential". Clarification is needed on whether the preceding limitation is related to Sequential Recommendation.

b)	The term PFCR appears in the abstract, but it is not introduced or explained before. It would be beneficial to provide a brief introduction or explanation for PFCR earlier.

c)	In Section 3.4.3, the explanation of why E can rectify the gradients is not clear. It seems that p-q is not an inverse operation of the clamp function stated in the Encryption Strategy section. A more detailed explanation is needed to clarify the reason behind the rectification of gradients.

d)	The specific prompt process is not clear, as it only involves the item embedding process. A more comprehensive explanation of the prompt process, including its components and steps, would enhance understanding.

3.	The figures are not well explained.

a)	Figure 1 lacks sufficient explanation. For instance, while the input is mentioned as the item text, it is not clear how the item text is transformed into a sequence, especially considering that the sequence encoder requires a sequence as input. A comprehensive explanation of the entire pipeline is recommended.

b)	Figure 2 lacks sufficient explanation. Specifically, the Id embedding table in the figure has not been introduced previously, raising questions about its role and origin. Additionally, it is unclear why the Sequence Encoder is frozen in the figure. A more detailed explanation is necessary to provide clarity.

4.	Some contents are contradictory. For example, in the first paragraph of Section 3.3, there is a statement that "the text encodings from different domains cannot naturally align in a unified semantic space." However, in the same paragraph, there is another statement suggesting that "it is intuitive to leverage the generality of natural language texts to bridge domain gaps." This appears contradictory and needs clarification to reconcile these statements and provide a coherent explanation.

5.	The experiments are not well explained.

a)	The reason for choosing the datasets is not clear.

b)	Some parameters, such as D and Mc in the section of 3.3.1 Discrete Item Code Learning, appear to be not tested, despite their apparent variability. Clarity is needed on whether these parameters are fixed or tested, and if not tested, an explanation would be beneficial.

c)	In the Hyper-parameters experiment, communication round 𝑡 and the number of quantified buckets 𝑏 are introduced without prior mention, and the reasons for conducting these experiments are not stated. Providing context and explanations for these hyper-parameters and the experimental setup would enhance the reader's understanding.

6.	There are some typos, such as in Table 2, where Recall@10 on the Office dataset shows a maximum value of 0.1213, but the bold formatting is applied to 0.1207.

7.	The format of the references is not unified. For example, the year information appears at the end of Reference 11, while it is missing in Reference 30. It is recommended to ensure consistency in the format of all references.

**Questions:**

Refer to the above comments.

**Reviewer Confidence:**

2: The reviewer is willing to defend the evaluation, but it is likely that the reviewer did not understand parts of the paper

**Scope:**

4: The work is relevant to the Web and to the track, and is of broad interest to the community

---

### Decision · Program_Chairs · 2024-01-22

**Decision:**

Accept

**Comment:**

The reviewers seem split as to the significance of this work. All reviewers agree that the authors have performed a rather extensive experimental evaluation. However some reviewers raise concerns related to some confusing parts of the paper, as well as to the interpretation of some (unexpected) results.